# Andrographis Reverses Gemcitabine Resistance through Regulation of ERBB3 and Calcium Signaling Pathway in Pancreatic Ductal Adenocarcinoma

**DOI:** 10.3390/biomedicines11010119

**Published:** 2023-01-03

**Authors:** Keisuke Okuno, Caiming Xu, Silvia Pascual-Sabater, Masanori Tokunaga, Tetsuji Takayama, Haiyong Han, Cristina Fillat, Yusuke Kinugasa, Ajay Goel

**Affiliations:** 1Department of Molecular Diagnostics and Experimental Therapeutics, Beckman Research Institute of City of Hope, Biomedical Research Center, Monrovia, CA 91016, USA; 2Department of Gastrointestinal Surgery, Tokyo Medical and Dental University, Tokyo 113-8510, Japan; 3Department of General Surgery, The First Affiliated Hospital of Dalian Medical University, Dalian 116004, China; 4Institut d’Investigacions Biomèdiques August Pi i Sunyer (IDIBAPS), 08036 Barcelona, Spain; 5Department of Gastroenterology and Oncology, Tokushima University Graduate School, Tokushima 770-8503, Japan; 6Molecular Medicine Division, The Translational Genomics Research Institute, Phoenix, AZ 85004, USA; 7City of Hope Comprehensive Cancer Center, Duarte, CA 91010, USA

**Keywords:** pancreatic ductal adenocarcinoma, Andrographis, gemcitabine, chemoresistance, ERBB3, calcium signaling pathway

## Abstract

Pancreatic ductal adenocarcinoma (PDAC) is one of the most lethal malignancies, primarily due to intrinsic or acquired resistance to chemotherapy, such as Gemcitabine (Gem). Naturally occurring botanicals, including Andrographis (Andro), can help enhance the anti-tumorigenic therapeutic efficacy of conventional chemotherapy through time-tested safety and cost-effectiveness. Accordingly, we hypothesized that Andro might reverse Gem resistance in PDAC. The critical regulatory pathways associated with Gem resistance in PDAC were identified by analyzing publicly available transcriptomic profiling and PDAC tissue specimens. A series of systematic in vitro experiments were performed using Gem-resistant (Gem-R) PDAC cells and patient-derived 3D-organoids to evaluate the Andro-mediated reversal of Gem resistance in PDAC. Transcriptomic profiling identified the calcium signaling pathway as a critical regulator of Gem-resistance (Fold enrichment: 2.8, *p* = 0.002). Within this pathway, high ERBB3 expression was significantly associated with poor prognosis in PDAC patients. The combination of Andro and Gem exhibited superior anti-cancer potential in Gem-R PDAC cells through potentiating cellular apoptosis. The combined treatment down-regulated ERBB3 and decreased intracellular calcium concentration in Gem-R PDAC cells. Finally, these findings were successfully interrogated in patient-derived 3D-organoids. In conclusion, we demonstrate novel evidence for Andro-mediated reversal of chemoresistance to Gem in PDAC cells through the regulation of ERBB3 and calcium signaling.

## 1. Introduction

Recent advances in treatment modalities have somewhat improved the prognosis of patients with pancreatic ductal adenocarcinoma (PDAC). However, this remains one of the most significant and lethal malignancies worldwide, with a dismal <8–10% 5-year overall survival (OS) rates [1,2,3]. One of the underlying reasons for this abysmal prognosis is the quick acquisition of resistance against chemotherapeutic drugs, such as Gemcitabine (Gem), which remain the standard of care for patients with PDAC [4]. Given increasing recognition that treatment regimens comprising multiple drugs might offer enhanced therapeutic efficacy by reducing the probability of developing chemoresistance [5,6], in the past decade, several clinical trials have examined the effectiveness of Gem-containing doublet regimens by combining it with other drugs (e.g., Oxaliplatin, Irinotecan, and Cisplatin), to improve overall prognosis of PDAC patients [7,8,9]. These exhaustive efforts have been disappointing but demonstrated that only nab-paclitaxel doublet treatment somewhat improves the OS vs. single-Gem regimen in PDAC [10]. While this may seem encouraging, the therapeutic benefits derived from combination treatments are often accompanied by increased simultaneous drug toxicity and added expense-which in part limits their clinical significance [11,12,13]. Nonetheless, these findings highlight the need to develop optimal combination therapies that are safe and cost-effective; and hence can have a significant impact on the management of patients suffering from this lethal malignancy.

Calcium signaling is a critical regulator of various cellular processes and is considered to play a significant role in the pathogenesis of multiple cancers, including PDAC [14,15,16]. In recent years, evidence indicates that calcium signaling significantly contributes to the acquisition of chemotherapeutic resistance in cancer cells. Surprisingly, this biological process has become an important therapeutic target in several cancers [15,16,17,18]. In PDAC, tissue-based, genomewide expression profiling studies have revealed that many essential genes within the “Response to calcium ion” pathway are frequently upregulated [19]. Furthermore, several previous studies have reported that calcium signaling also plays an integral role in Gem resistance [20,21]. In addition, calcium channels also are involved in multi-drug resistance [22], and calcium channel blockers, such as amlodipine, verapamil, and RP4010, could help overcome chemotherapeutic resistance in PDAC cells [23,24,25]. Accordingly, calcium signaling was recently considered one of the attractive therapeutic targets for reversing chemoresistance in PDAC.

Andrographolide, a labdane diterpenoid isolated from the traditional herb Andrographis paniculate [26], is known to have a strong calcium channel-blocking capacity [27,28,29,30,31]. Furthermore, in the last two years, research from our team and other laboratories has demonstrated the exciting potential of Andrographis (Andro) as an anti-cancer compound by its ability to regulate multiple oncogenic pathways in various cancers [32,33,34,35,36,37], as well as its ability to augment the anti-cancer therapeutic efficacy of other chemotherapeutic drugs [33,34,35,36,37]. These findings are very promising considering the anti-cancer efficacy of such naturally occurring dietary botanicals together with their safety and cost-effectiveness, such an adjunctive use of such botanicals in combination with various chemotherapeutic drugs might help overcome the clinical challenge for the treatment of patients with PDAC more effectively [38,39]. Given this evidence, we hypothesized that Andro might facilitate the reversal of Gem resistance in PDAC cells by regulating the calcium signaling pathway to exert a superior anti-cancer activity in this fatal malignancy.

Herein, in this study, by analyzing genomewide transcriptomic profiling data, followed by gene expression analyses in clinical specimens from patients with PDAC, we demonstrated that the calcium signaling pathway plays a critical role in reversing Gem resistance in this disease. Furthermore, to better understand the molecular events underpinning the Andro-mediated sensitization of Gem resistance in PDAC, a series of experiments using multiple Gem-resistant PDAC cell lines was performed, followed by confirmation of these findings in patient-derived 3D-organoids.

## 2. Materials and Methods

### 2.1. Patient Cohorts

This study included an analysis of 345 PDAC patients from multiple cohorts, including two public datasets, The Cancer Genome Atlas (TCGA; n = 178), GSE71729 (n = 125), and a clinical cohort (n = 37). TCGA data were downloaded from the University of California Santa Cruz Xena Browser (https://xenabrowser.net/ [accessed on 23 November 2022]) and the GSE71729 dataset from the Gene Expression Omnibus (https://www.ncbi.nlm.nih.gov/geo/ [accessed on 23 November 2022]). The patient’s characteristics of the public dataset are shown in Appendix A.

In the clinical cohort, formalin-fixed paraffin-embedded (FFPE) tissue specimens were examined, which were obtained from 37 PDAC patients enrolled at Tokushima University, Japan. All patients were initially treated with Gem-containing chemotherapy-Gem + nab-paclitaxel or Gem + S-1 (Tegafur/Gimeracil/Oteracil), and all FFPE tissues were obtained by endoscopic ultra-sound-fine needle aspiration (EUS-FNA), before initiation of treatment. All tumors were histologically diagnosed as PDAC and were classified according to the Union for International Cancer Control (UICC) TNM classification of Malignant Tumors version 7. Patients were classified as responders (complete response [CR], partial response [PR], and stable disease [SD]) and non-responders (progressive disease [PD]) based on the best response evaluated by the Response Evaluation Criteria in Solid Tumors (RECIST) version 1.1 [40]. The clinicopathological characteristics of the clinical cohort are shown in Appendix A. The study was conducted in compliance with the Declaration of Helsinki and was approved by the Institutional Review Boards of all institutions. Written informed consent was obtained from all patients.

### 2.2. Genomewide Expression Profiling and Pathway Enrichment Analysis in Gemcitabine-Resistant PDAC Cells

To discover differentially regulated genes in Gemcitabine-resistant (Gem-R) PDAC cells, the gene expression profiling patterns were compared between parental and Gem-R PDAC cells in two publicly available datasets (GSE148200 and GSE140077), which were downloaded from the Gene Expression Omnibus database (https://www.ncbi.nlm.nih.gov/geo/ [accessed on 23 November 2022]). A given gene was differentially regulated when it exhibited a Log_2_ fold-change (FC) > ±1.0 and a *p* value of < 0.01. The data were analyzed using the Kyoto Encyclopedia of Genes and Genomes (KEGG) pathway analysis tools. Enrichment pathway analysis was carried out using DAVID bioinformatic database 2021 (https://david.ncifcrf.gov/ [accessed on 23 November 2022]) [41], and a pathway was considered enriched when it had the fold enrichment > 2.0 and *p* < 0.05. The scatter plots and heatmap of differentially regulated genes are generated based on the results of enrichment pathway analysis by the DAVID bioinformatic database.

### 2.3. Cell Culture and Reagents

The human PDAC cell lines (MIA PaCa-2 and BxPC-3) were obtained from the American Type Culture Collection (ATCC; Manassas, VA, USA). All cell lines were cultured in Roswell Park Memorial Institute medium from Gibco, Carlsbad, CA, USA, and were supplemented with 10% fetal bovine serum (Gibco) and 1% penicillin-streptomycin (Gibco). The cells were maintained in a humidified incubator (37 °C, 5% CO_2_) and harvested with 0.05% trypsin-0.03% EDTA (Invitrogen, Carlsbad, CA, USA). All cell lines were tested for mycoplasma regularly and authenticated using a panel of genetic and epigenetic markers every 4–6 months. Andrographis (standardized to 20% andrographolide content; EuroPharma, Green Bay, WI, USA) and Gemcitabine (Sigma-Aldrich, St. Louis, MO, USA) were both dissolved in the dimethyl sulfoxide (DMSO) to prepare stock concentrations. The stock solution of Andro and Gem was stored at −20 °C and diluted to appropriate experimental concentrations in phosphate-buffered solution (PBS) before use. Gem-R PDAC cells-Gem-R MIA PaCa-2 and Gem-R BxPC-3-were established by continuously culturing cells with increasing doses of Gem, as described in a previous study [42]. 

### 2.4. Total RNA Extraction and Real-Time Quantitative Reverse Transcription Polymerase Chain Reaction

Total RNA extraction from FFPE specimens and PDAC cells was performed using an AllPrep DNA/RNA FFPE kit (Qiagen, Hilden, Germany) and an ALLPrep DNA/RNA/miRNA Universal kit (Qiagen), respectively, as described previously [43,44,45]. Total RNA was reverse-transcribed to complementary DNA (cDNA) using a high-capacity cDNA Reverse Transcription Kit (Thermo Fischer Scientific, Waltham, MA, USA). The quantitative reverse transcription polymerase chain reaction (qRT-PCR) assays were performed using the QuantStudio 6 Flex RT-PCR System (Applied Biosystems, Foster City, CA, USA) and a SensiFAST SYBR Lo-ROX Kit (Bioline, London, UK). The expression of target genes was quantified by the ΔΔCt method normalized against the β-actin gene [46]. The primers used in this study are described in Appendix A.

### 2.5. Cell Viability, Colony Formation, and Wound Healing Assays

The cell viability, colony formation, and wound healing assays were performed as described previously [34,35,36,47,48,49,50,51,52]. For cell viability assays, 5 × 10^3^ cells per well in 96-well plates were incubated with various concentrations of Andro, Gem, and their combination for 48 h. Uniform DMSO concentrations were used between each treatment group. The cell viability was measured using the Cell Counting Kit-8 (CCK-8) solution (Dojindo, Kumamoto, Japan) and a microplate reader (Tecan Trading AG, Männedorf, Switzerland). To assess Andro’s reversal of chemoresistance to Gem, the combination index (CI) was calculated using the Chou-Talalay equation [53]. For colony formation assays, 5 × 10^2^ cells were seeded per well in 6-well plates, followed by treatment with Andro, Gem, and their combination for 48 h, and colony formation allowed for one week. After that, the number of colonies was counted by 1% crystal violet staining using the Image-J 1.53 s software (http://imagej.nih.gov/ij/index.html [accessed on 23 November 2022]). Cell monolayers were scratched with a 200-μL pipette tip for wound healing assays and treated with Andro, Gem, and their combination. Cell migration was observed for up to 24 h, and the percentage of wound closure was calculated using the Image-J 1.53 s software. All experiments were performed in triplicates.

### 2.6. Annexin V Binding Assays

Apoptosis assays were performed using Muse Annexin V & Dead Cell Reagent (Luminex Corp, Austin, TX, USA) and Muse Cell Analyzer (EMD Millipore Corp, Hayward, CA, USA), as described previously [44]. The assay utilizes Annexin V to detect phosphatidylserine on the external membrane of apoptotic cells. After treatment, cells were harvested, and 100 μL of cell suspension was added to 100 μL of Muse Annexin V & Dead Cell Reagent. The apoptotic cell fraction was measured using the Muse Cell Analyzer. 

### 2.7. Protein Extraction and Western Immunoblotting

Total protein extraction and western immunoblotting were performed as described previously [54,55]. Total proteins were extracted from cells using RIPA Lysis and Extraction Buffer containing a protease inhibitor cocktail (Thermo Fischer Scientific). Proteins were separated by electrophoresis in a 10% Mini-PROTEAN TGX^TM^ Precast Gel (BIO-RAD, Hercules, CA, USA) and electro-transferred onto nitrocellulose membranes. Membranes were incubated overnight with diluted primary antibodies. Primary antibodies against Bcl-2 (1:1000, #15071; Cell Signaling Technology, Danvers, MA, USA), Caspase-3 and cleaved Caspase-3 (1:1000, #9662; Cell Signaling Technology), and Cyclin-D1 (1:1000, #2978; Cell Signaling Technology) were used. After that, membranes were incubated with secondary antibodies (#7074 or #7076; Cell Signaling Technology) for 1 h. Immunoblots were visualized using an HRP-based chemiluminescence kit (Thermo Fisher Scientific). β-actin (1:1000, #58169; Cell Signaling Technology) was used as an internal control, and the relative protein levels were quantified using the Image-J 1.53 s software.

### 2.8. Fluo-4-Based Calcium Imaging

The transient changes in intracellular calcium concentrations were determined using Fluo-4 Calcium Imaging Kit (F10489; Thermo Fisher Scientific). The cells were treated with Andro, Gem, and their combination for 48 h and washed with PBS. After that, cells were incubated with Fluo-4 for 1 h and imaged using 494 nm excitation and 506 nm emission in a microplate reader (Tecan Trading AG). Cells were also observed under 160× magnification using a fluorescent microscope.

### 2.9. Patient-Derived Tumor 3D-Organoids

Tumor 3D-organoids from patients with PDAC were generated as described previously [56]. Patients were anonymously coded in compliance with the Declaration of Helsinki, and written informed consent was obtained from all patients with approval from the ethics committees of the institution. The tumor organoids were suspended in a 24-well plate such that they formed a dome in 40 μL of Matrigel (Corning, Tehama County, CA, USA) with 500 μL of PancreaCult^TM^ Organoid Growth Medium (100-0781, STEMCELL Technologies, Vancouver, BC, Canada) containing epidermal growth factor (STEMCELL Technologies). The 3D-organoids were treated with appropriate concentrations of Andro, Gem, and their combination. The control organoids were treated with a very low concentration of DMSO. Tumor organoids about 50 microns in diameter were observed under a bright-field microscope (magnification 100×). The number and size of organoids were measured using the Image-J 1.53 s software.

### 2.10. Statistical Analysis

All data were expressed as mean ± standard deviation (SD), and differences between continuous values of each group were analyzed using a two-sided Student’s *t*-test. Survival curves were conducted using the Kaplan–Meier method by using the log-rank test. The factors acquired from univariate analysis (*p* < 0.10) were included in multivariate Cox regression analysis. Spearman rank correlation coefficient was indicated as *R* values. All statistical analyses were conducted using EZR version 1.55, a graphical user interface for R (R Foundation for Statistical Computing, Vienna, Austria, version 4.1.2) designed to add statistical functions and is frequently used in biostatistics [57]. All *p* values were two-sided, and *p* < 0.05 was considered statistically significant. 

## 3. Results

### 3.1. The Calcium Signaling Pathway Significantly Correlates with Gemcitabine Resistance in PDAC Cells

First, we undertook a comprehensive genomewide transcriptomic profiling analysis in parental and Gem-R PDAC cells from publicly available datasets (GSE148200 and GSE140077) to identify the most critical signaling and regulatory pathway correlated with Gem resistance. In this analysis, we discovered that 2420 and 3971 differentially regulated genes (log_2_FC > ±1.0, *p* < 0.01) in MIA PaCa-2 and BxPC-3 cell lines, respectively (Figure 1A), while 415 genes were shared between both these PDAC cell lines (Figure 1B). When we performed pathway enrichment analyses on these 415 genes using DAVID bioinformatic database 2021 (https://david.ncifcrf.gov/ [accessed on 23 November 2022]) [41], 14 KEGG pathways were found to be preferentially enriched (Fold enrichment > 2.0, *p* < 0.05; Figure 1A,C). Among these, calcium signaling emerged as one of the most relevant signaling pathways because it has also received considerable attention in recent years for its role in mediating chemotherapeutic resistance in various cancers [15,16,17,18]. Therefore, we focused further attention on exploring the role of this pathway for Gem resistance in PDAC cells (Fold enrichment: 2.8, *p* = 0.002; Figure 1C) and noted that 13 genes were differentially regulated in PDAC cells with and without resistance to Gem (Figure 1D). Taken together, these results suggested that the calcium signaling pathway might play a significant role in Gem resistance in PDAC cells, and we selected this pathway for subsequent experiments.

### 3.2. High ERBB3 Expression Associates with Poor Survival Outcomes and Gemcitabine Resistance in PDAC Patients

Among the 13 differentially regulated genes in the calcium signaling pathway, ERBB3, a member of the epidermal growth factor receptor, is considered to have a critical oncogenic role in pancreatic cancer pathogenesis, disease progression, and acquisition of chemoresistance [58,59,60] and more importantly, has recently gained increased attention as a potential therapeutic target in PDAC [60,61]. Based on these findings, we focused on ERBB3 and assessed its impact on survival and prognosis in PDAC patients. When we analyzed two public datasets (TCGA and GSE71729), wherein we observed that the OS was significantly reduced in the patient group with high ERBB3 expression in both TCGA (3-year OS: 46.8% vs. 26.9%; *p* = 0.02; Figure 2A) and GSE71729 (3-year OS: 27.0% vs. 16.1%; *p* = 0.02; Figure 2B) datasets. Furthermore, in multivariate Cox regression analysis for OS in the TCGA dataset, ERBB3 level was a significant independent prognostic factor of OS (hazard ratio [HR]: 1.59; 95% confidence interval [CI]: 1.03–2.47; *p* = 0.04; Figure 2C). Overall, these results highlighted that high ERBB3 expression was significantly associated with poor prognosis in PDAC patients.

Next, to investigate the impact of ERBB3 on Gem resistance in PDAC patients, we performed qRT-PCR assays in clinical tissue specimens from patients with unresectable cancers obtained by EUS-FNA before initiation of treatment. All PDAC patients in this cohort were initially treated with Gem-containing chemotherapy—35 patients were treated with Gem + nab-paclitaxel, and two patients were Gem + S-1. This cohort included 23 responders and 14 non-responders based on RECIST guidelines (Appendix A). Of note, non-responders exhibited a significantly higher ERBB3 level compared to responders (*p* = 0.02; Figure 2D). In addition, patients with high tumoral ERBB3 expression demonstrated substantially poorer progression-free survival (PFS) compared to those with low ERBB3 expression (median PFS: 0.56 vs. 0.31 years; *p* = 0.02; Figure 2E). More interestingly, the ERBB3 expression level confirmed a negative correlation with PFS time (*R* = −0.41; *p* = 0.01; Figure 2F). Collectively, high ERBB3 expression emerged as a significant prognostic indicator for patient survival as well as Gem resistance in PDAC. 

### 3.3. Andrographis Potentiates the Chemosensitivity to Gemcitabine in Gemcitabine-Resistant PDAC Cells

Andrographis is well-known as a potent calcium channel blocker [27,28,29,30,31]. Our recent studies investigated Andro’s explicit anti-cancer activity, individually and in combination with various chemotherapeutic drugs and other naturally occurring botanicals [32,33,34,35,36,37]. Given this evidence, we hypothesized that Andro might be able to reverse Gem resistance by regulating calcium signaling pathways in PDAC. To test this hypothesis, the cell viability assays were performed following treatment with Andro, Gem, and their combination in Gem-R PDAC cells (Gem-R MIA PaCa-2 and Gem-R BxPC-3), which were established previously (Appendix A) [42]. In these assays, the IC_50_ of the combination treatment was Andro: 22.1 μg/mL, Gem: 221 nM in Gem-R MIA PaCa-2 cells, and Andro: 26.2 μg/mL, Gem: 261 nM in Gem-R BxPC-3 cells (Figure 3A). Therefore, all experiments were performed at a concentration of Andro: 25 μg/mL, Gem: 250 nM, which is approximately IC_50_ of combination treatment in both Gem-R PDAC cells. To assess the ability of Ando to enhance chemosensitivity to Gem, the CI was calculated using the Chou-Talalay equation [53]. The CI value at 50% inhibitory concentration in Gem-R MIA PaCa-2 and BxPC-3 were 0.62 and 0.74, respectively, with less than 1.0 value, which was conceived as a synergistic interaction (Figure 3B). 

Next, Andro’s chemosensitizing properties were assessed in colony formation and wound healing assays. With regard to the colony formation assays, the combination of Andro and Gem significantly reduced cellular clonogenicity compared to either treatment individually (*p* = 0.27 vs. Andro; *p* = 0.02 vs. Gem in Gem-R MIA PaCa-2; *p* = 0.02 vs. Andro; *p* < 0.01 vs. Gem in Gem-R BxPC-3; Figure 3C). Similarly, the wound healing assays revealed that the combined treatment resulted in significant inhibition of migration in Gem-R MIA PaCa-2 (FC = 0.58 vs. Andro, *p* = 0.03; FC = 0.38 vs. Gem, *p* < 0.01) and Gem-R BxPC-3 cells (FC = 0.42 vs. Andro, *p* = 0.03; FC = 0.31 vs. Gem, *p* < 0.05; Figure 3D). Collectively, these data confirmed that Andro significantly enhanced the chemosensitivity to Gem and even reversed Gem resistance in Gem-R PDAC cells.

### 3.4. The Combination of Andrographis and Gemcitabine Promotes Cellular Apoptosis in Gemcitabine-Resistant PDAC Cells

To clarify the apoptotic modulation in the Andro-mediated reversal of Gem resistance in PDAC cells, Annexin V binding assays were performed with a combination treatment of Andro and Gem in Gem-R PDAC cells. The Annexin V binding assays demonstrated that the combined treatment with Andro and Gem enhanced the apoptotic potential compared to either treatment individually in both Gem-R MIA PaCa-2 (Andro vs. Combination, 20.3% vs. 31.4%, *p* = 0.04; Gem vs. Combination, 10.9% vs. 31.4%, *p* = 0.03) and Gem-R BxPC-3 (Andro vs. Combination, 11.3% vs. 16.1%, *p* = 0.16; Gem vs. Combination, 6.8% vs. 16.1%, *p* = 0.03; Figure 4A). 

Furthermore, when the expression of apoptosis-related genes was evaluated by qRT-PCR and western immunoblotting assays, the combined treatment significantly down-regulated the expression of Bcl-2 at both the mRNA (*p* = 0.04 vs. Andro; *p* = 0.03 vs. Gem in Gem-R MIA PaCa-2; *p* = 0.12 vs. Andro; *p* = 0.04 vs. Gem in Gem-R BxPC-3; Figure 4B) and protein levels (*p* = 0.01 vs. Andro; *p* = 0.04 vs. Gem in Gem-R MIA PaCa-2; *p* = 0.02 vs. Andro; *p* = 0.02 vs. Gem in Gem-R BxPC-3; Figure 4C). In addition, Cyclin-D1 was significantly down-regulated by the combined treatment at mRNA (*p* = 0.01 vs. Andro; *p* < 0.05 vs. Gem in Gem-R MIA PaCa-2; *p* = 0.01 vs. Andro; *p* < 0.01 vs. Gem in Gem-R BxPC-3; Figure 4B) and protein levels (*p* < 0.01 vs. Andro; *p* = 0.02 vs. Gem in Gem-R MIA PaCa-2; *p* = 0.10 vs. Andro; *p* = 0.01 vs. Gem in Gem-R BxPC-3; Figure 4C). The combined treatment significantly up-regulated the expression of Caspase-3 at the mRNA (*p* = 0.03 vs. Andro; *p* = 0.02 vs. Gem in Gem-R MIA PaCa-2; *p* = 0.02 vs. Andro; *p* < 0.01 vs. Gem in Gem-R BxPC-3; Figure 4B) and protein levels (*p* < 0.01 vs. Andro; *p* < 0.01 vs. Gem in Gem-R MIA PaCa-2; *p* < 0.01 vs. Andro; *p* < 0.01 vs. Gem in Gem-R BxPC-3; Figure 4C), while cleaved Caspase-3 was significantly up-regulated by the combined treatment (*p* < 0.01 vs. Andro; *p* < 0.01 vs. Gem in Gem-R MIA PaCa-2; *p* = 0.15 vs. Andro; *p* = 0.04 vs. Gem in Gem-R BxPC-3; Figure 4C). Consistent with the results from the cell viability assays, Andro treatment significantly improved the anti-cancer potential by promoting cellular apoptosis in Gem-R PDAC cells.

### 3.5. The Combination of Andrographis and Gemcitabine Decreases Intracellular Calcium Concentration in Gemcitabine-Resistant PDAC Cells

To examine whether the combined treatment with Andro and Gem targets the calcium signaling pathway in Gem-R PDAC cells, the expression of ERBB3 was evaluated by qRT-PCR assays in the combination treatment group. Compared to the untreated group, Andro significantly down-regulated ERBB3 expression in both cell lines (FC = 0.17, *p* = 0.03 in Gem-R MIA PaCa-2; FC = 0.52, *p* = 0.01 in Gem-R BxPC-3; Figure 5A), while the combined treatment further down-regulated ERBB3 in both Gem-R MIA PaCa-2 (FC = 0.33 vs. Andro, *p* < 0.05; FC = 0.05 vs. Gem, *p* < 0.01) and Gem-R BxPC-3 cells (FC = 0.35 vs. Andro, *p* < 0.05; FC = 0.19 vs. Gem, *p* = 0.03; Figure 5A). 

Next, we assessed the intracellular calcium concentrations with combination treatment using Fluo-4-based calcium imaging assays in the microplate reader and fluorescent microscope. These assays demonstrated that intracellular calcium concentration was significantly decreased by the combined treatment compared to either treatment individually in both Gem-R MIA PaCa-2 (*p* < 0.01 vs. Andro; *p* < 0.01 vs. Gem) and Gem-R BxPC-3 cells (*p* < 0.01 vs. Andro; *p* < 0.01 vs. Gem; Figure 5B,C). These results suggested that calcium signaling is one of the key regulators in the Andro-mediated reversal of Gem resistance in PDAC cells.

### 3.6. The Combined Treatment with Andrographis and Gemcitabine Effectively Enhances Anti-Cancer Activity in PDAC Patient-Derived 3D-Organoid Models

To validate our cell culture-based findings in the 3D cultures, the anti-cancer potential of Andro and Gem, individually or combined, was evaluated in tumor-derived 3D-organoid models generated from PDAC patients [56]. For the organoid model experiments, the combined treatment with Andro and Gem significantly inhibited the formation of patient-derived 3D-organoids compared to either treatment individually in IDIT5 (FC = 0.46 vs. Andro, *p* < 0.01; FC = 0.44 vs. Gem, *p* = 0.04), IDIT6 (FC = 0.62 vs. Andro, *p* = 0.09; FC = 0.49 vs. Gem, *p* < 0.01), and IDIT7 (FC = 0.73 vs. Andro, *p* = 0.03; FC = 0.69 vs. Gem, *p* = 0.07; Figure 6A,B). In addition, the growth of patient-derived 3D-organoids was also significantly hampered by the combination treatment vs. either treatment individually in IDIT5 (*p* < 0.01 vs. Andro; *p* = 0.01 vs. Gem), IDIT6 (*p* < 0.01 vs. Andro; *p* < 0.01 vs. Gem), and IDIT7 organoids (*p* < 0.01 vs. Andro; *p* = 0.03 vs. Gem; Figure 6A,C). Overall, Andro could significantly enhance the anti-cancer activity of Gem in inhibiting patient-derived tumor organoids, and our cell culture-based findings were successfully validated in 3D cultures.

## 4. Discussion

Developing newer and optimal combinations of safe and cost-effective therapeutic modalities for effective and improved treatment of a fatal malignancy such as PDAC is an unmet major clinical challenge. Naturally occurring dietary botanicals are frequently used as supplementary adjunctive therapies in various cancers, including PDAC [38,62], primarily because of their cost-effectiveness with time-tested safety through anecdotal use in traditional systems of medicines. Additionally, current research trends in the field suggest that a combination of conventional chemotherapeutic drugs together with dietary botanicals can help enhance the overall anti-cancer therapeutic efficacy in various cancers [63,64,65,66,67,68]. Consequently, several interventional clinical trials have been recently conducted in patients with advanced PDAC using the combination of standard chemotherapy and a variety of botanicals [62,69]—e.g., NCT00192842, NCT00486460 (Gem with Curcumin), NCT02336087 (Gem, nab-paclitaxel, metformin with Curcumin-containing supplement), NCT02948309 (standard chemotherapy with Mistletoe extract). In this regard, the present study interrogated whether an optimal combination of a chemotherapeutic drug and a naturally occurring dietary botanical—Gemcitabine and Andrographis, could collectively enhance the chemosensitivity to Gem and reverse Gem resistance in PDAC cells.

To overcome the chemotherapeutic resistance, naturally occurring dietary botanicals have been recently well-studied for their adjunctive use in combination with various chemotherapeutic drugs in the field of cancers, including PDAC [38,39]. In this regard, Andro has garnered increased attention because of its chemosensitizing potential to other drugs—e.g., Andro enhanced chemosensitivity of various cancer cells to Cisplatin and Doxorubicin [70,71]. Additionally, Andro reversed Cisplatin resistance in ovarian cancer by enhancing cellular apoptosis [72]. Furthermore, in our previous studies about colorectal cancer, Andro reversed 5-FU resistance through suppression of DKK1 and β-catenin/Wnt-signaling [33,34]. Consistent with these previous studies, in the present study, we first demonstrated the Andro-mediated Gem resistance reversal by using a series of cell-culture experiments with patient-derived 3D-organoids in PDAC. Taken together, our findings suggest a potential combinatorial adjunctive therapy along with current chemotherapeutic regimens or an alternative treatment for patients who struggle to receive current chemotherapeutic therapies for such a lethal malignancy.

Multiple signaling pathways contribute to the intrinsic and acquired resistance to Gemcitabine in PDAC and may serve as potential therapeutic targets. Since activating mutations within the KRAS gene are observed in 90% of PDAC patients [73,74], tyrosine kinase-related signaling pathways, including MAPK, PI3K-Akt, JAK-STAT, and NRF2, have garnered close attention as potential therapeutic targets in PDAC [75,76,77]. In fact, in the present study, our pathway analyses confirm this hypothesis, as we identified that Gem resistance is associated with several tyrosine kinase-related pathways, including MAPK and PI3K-Akt signaling pathways (Figure 1C). In this regard, recent evidence has shown that calcium signaling remarkably contributes to chemotherapeutic resistance in cancer cells. Targeting the calcium signaling pathway has emerged as a significant perspective in various cancers, including Gem resistance in PDAC [15,16,17,18,20,21,22]. Within this context, ERBB3, a calcium signaling pathway gene, has gained increased attention as a potential therapeutic target against PDAC [60,61]. From a functional standpoint, it has been shown that the transmembrane region of ERBB3 confers a ligand-dependent calcium influx [78,79], and loss or down-regulation of ERBB3 in cancer cells results in cellular apoptosis [80,81]. In line with this evidence, our transcriptomic profiling analyses in Gem sensitive and resistant cells demonstrated that calcium signaling was one of the critical regulators for Gem resistance in PDAC cells. In addition, based on our clinical cohort analysis, high ERBB3 expression was significantly correlated with poor prognosis and Gem resistance in PDAC patients. In a series of cell culture studies using Gem-R PDAC cells, Andro down-regulated the expression of ERBB3 and decreased intracellular calcium concentrations. Our results suggested that Andro inhibited ERBB3′s ligand-dependent calcium influx and decreased intracellular calcium concentration in Gem-R PDAC cells, which resulted in increased cellular apoptosis and reversal of Gem resistance (Figure 6D). 

In recent decades, Andro’s potential anti-cancer activity was well-studied in multiple cancers. In this regard, our research team has also revealed the potential of Andro as an anti-cancer compound by its ability to regulate various oncogenic pathways, including ferroptosis, autophagy, β-catenin/Wnt-signaling and DNA replication [32,33,34,35,36,37]. In addition, Andro can inhibit key cancer-related pathways, such as PI3K-Akt signaling and NF-kB pathways, and interfere with the immune mechanism by impeding T-cell activation [82,83,84]. Furthermore, interestingly, several previous studies demonstrated Andro’s biological properties in RAS-transformed cancer cells [85,86], suggesting Andro’s promising potential against PDAC, where more than 90% have mutations of KRAS [73,74]. 

We would like to acknowledge some of the limitations of this study. First, it primarily consisted of just two Gem-R PDAC cell lines and tumor-derived 3D-organoids from three patients. Accordingly, future studies are warranted regarding the Andro-mediated reversal of Gem resistance in PDAC cells, including validation of our findings in the other Gem-R PDAC cell lines and Gem-R tumor-derived 3D-organoids. Second, because this study did not include an in vivo experimental system, Andro’s possible dose and potential toxicity were not examined. The dose of Andro (25 μg/mL) in this study was similar to the one used in our previous studies (15–40 μg/mL) [32,33,34,35,36,37], where the mice treated with Andro exhibited a robust anti-cancer activity with no adverse events. From these findings, we believe our dose might be acceptable; however, further studies are warranted to determine a more precise dose for clinical use. Third, our pathway analyses did not include data about treatment with the combination of Andro and Gem. However, since the primary purpose of the present study was to evaluate Andro’s anti-cancer potential to enhance chemosensitivity to Gem in PDAC cells, this might not be a significant limitation. Accordingly, we focused on the calcium signaling pathway, which was identified as one of the critical regulators of Gem resistance in our transcriptomic profiling and pathway enrichment analyses. Whereas RNA sequencing in PDAC cells treated with Andro and combination might enable the identification of more specific pathways, future studies with transcriptomic profiling in combination treatment might reveal unknown mechanisms in Gem-R PDAC cells. Fourth, this study did not include a comparison with other drugs (e.g., calcium channel blockers) and compounds. Accordingly, future studies are warranted to investigate the detailed molecular mechanisms of the Andro-mediated Gem resistance reversal, including the comparison with calcium channel blockers (e.g., verapamil) and other botanicals.

## 5. Conclusions

We first investigated the Andro-mediated Gem resistance reversal in PDAC cells using a systematic series of Gem resistant cell cultures and patient-derived tumor organoid models. Calcium signaling was one of the critical regulators of Gem resistance in PDAC, and Andro could overcome Gem resistance via the regulation of ERBB3 and calcium signaling. Our findings could provide essential evidence of the combined treatment with Andro and Gem as a new safe, cost-effective therapeutic modality.

## Figures and Tables

**Figure 1 biomedicines-11-00119-f001:**
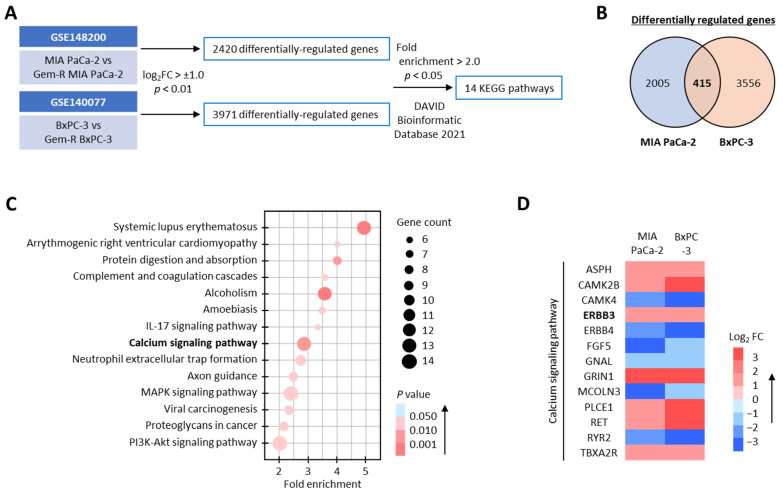
Calcium signaling pathway correlates with Gemcitabine resistance in PDAC cells. (**A**) Schematic illustration of the differentially regulated genes and pathway enrichment discovery analyses in PDAC cells using GSE148200 and GSE140077 datasets. A gene was differentially regulated when it had Log_2_ FC > ±1.0, and *p* < 0.01, and a pathway was considered enriched when it had fold enrichment > 2.0 and *p* < 0.05. (**B**) Venn-diagram of differentially regulated genes in PDAC cells. (**C**) Scatter plots of KEGG pathway enrichment analysis of differentially regulated genes in Gem-R PDAC cells. The circle area indicates the number of differentially regulated genes in the pathway, and the circle color represents the *p* value range. (**D**) Heatmap of differentially regulated genes in the calcium signaling pathway. PDAC, pancreatic ductal adenocarcinoma; Gem-R, Gemcitabine-resistant; FC, fold-change; KEGG, Kyoto Encyclopedia of Genes and Genomes.

**Figure 2 biomedicines-11-00119-f002:**
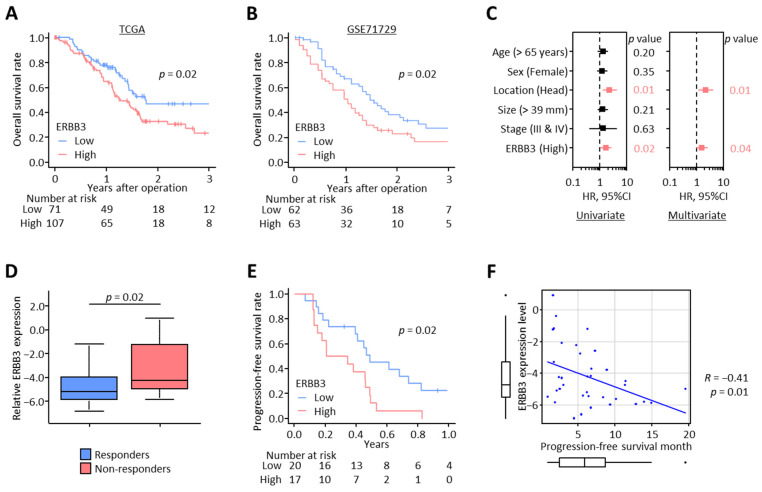
High ERBB3 expression was associated with poor survival outcomes and Gemcitabine resistance in PDAC patients. (**A**,**B**) Kaplan–Meier curves depicting the overall survival for PDAC patients with high or low ERBB3 expression in TCGA (**A**) and GSE71729 (**B**). (**C**) Forest plots with HR for each key clinical characteristic and ERBB3 expression level in univariate and multivariate Cox regression analysis for overall survival in TCGA. (**D**) Box plots representing the ERBB3 expression for responders or non-responders to Gem-containing chemotherapy in a clinical cohort. (**E**) In a clinical cohort, Kaplan–Meier curves of the progression-free survival in patients treated with Gem-containing chemotherapy with high or low ERBB3 expression. (**F**) Pearson correlation of ERBB3 expression level and progression-free survival time in PDAC patients treated with Gem-containing chemotherapy (*R* = −0.41, *p* = 0.01). PDAC, pancreatic ductal adenocarcinoma; TCGA, The Cancer Genome Atlas; HR, hazard ratio; CI, confidence interval; Gem, Gemcitabine.

**Figure 3 biomedicines-11-00119-f003:**
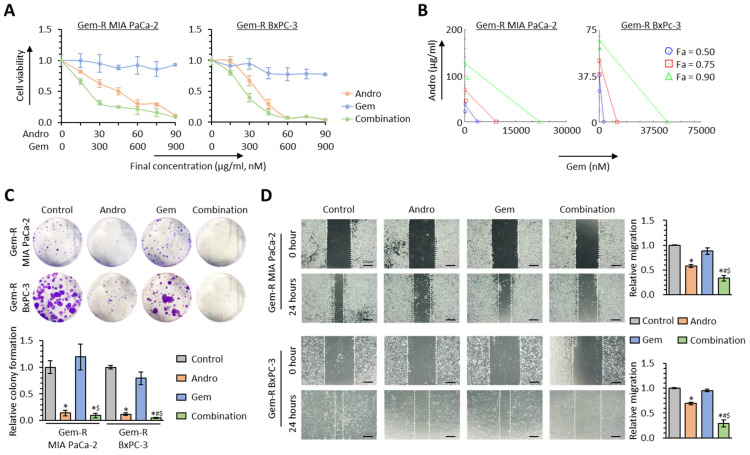
Andrographis potentiates the chemosensitivity to Gemcitabine in resistant PDAC cells. (**A**) Cell viability assays following treatment with Andro, Gem, and their combination for 48 h in Gem-R MIA PaCa-2 and BxPC-3. Error bars are the mean ± SD. (**B**) Isobologram analyses after combined treatment with Andro and Gem in Gem-R MIA PaCa-2 and BxPC-3. (**C**) Colony formation assays to evaluate clonogenicity of Gem-R PDAC cells following treatment with Andro, Gem, and their combination. The average (column) ± SD is indicated (* *p* < 0.05 vs. control, ^#^
*p* < 0.05 vs. Andro, ^$^
*p* < 0.05 vs. Gem). (**D**) Wound healing assay following treatment with Andro, Gem, and their combination for 24 h in Gem-R PDAC cells. Photographs show representative scratched and wound-recovering areas (marked by white lines). Scale bar = 250 μm. The average (column) ± SD is indicated (* *p* < 0.05 vs. control, ^#^
*p* < 0.05 vs. Andro, ^$^
*p* < 0.05 vs. Gem). PDAC, pancreatic ductal adenocarcinoma; Gem-R, Gemcitabine-resistant; Andro, Andrographis; Gem, Gemcitabine; SD, standard deviation.

**Figure 4 biomedicines-11-00119-f004:**
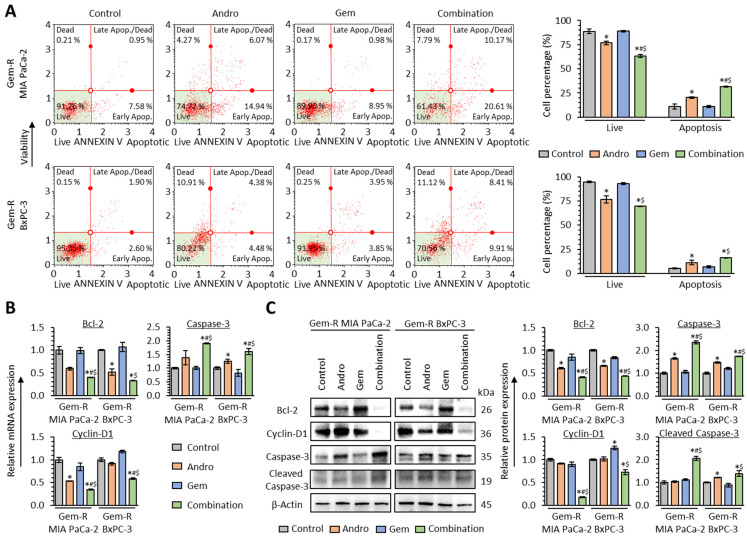
Andrographis and Gemcitabine enhance cellular apoptosis in Gemcitabine-resistant PDAC cells. (**A**) Representative images of cells undergoing apoptosis in Annexin V binding assay following treatment with Andro, Gem, and their combination for 48 h in Gem-R PDAC cells. The average ratio (column) ± SD of live and apoptotic cells is indicated (* *p* < 0.05 vs. control, ^#^
*p* < 0.05 vs. Andro, ^$^
*p* < 0.05 vs. Gem). (**B**) qRT-PCR analyses of apoptosis-related genes (Bcl-2, Cyclin-D1, and Caspase-3) in Gem-R PDAC cells following treatment with Andro, Gem, and their combination for 48 h. Relative expression was calculated using β-Actin mRNA expression as an internal control. The average (column) ± SD is indicated (* *p* < 0.05 vs. control, ^#^
*p* < 0.05 vs. Andro, ^$^
*p* < 0.05 vs. Gem). (**C**) Western immunoblotting of apoptosis-related genes (Bcl-2, Cyclin-D1, Caspase-3, and cleaved Caspase-3) in Gem-R PDAC cells following treatment with Andro, Gem, and their combination for 48 h. β-Actin protein was used as an internal control. The average (column) ± SD is indicated (* *p* < 0.05 vs. control, ^#^
*p* < 0.05 vs. Andro, ^$^
*p* < 0.05 vs. Gem). PDAC, pancreatic ductal adenocarcinoma; Gem-R, Gemcitabine-resistant; Andro, Andrographis; Gem, Gemcitabine; SD, standard deviation.

**Figure 5 biomedicines-11-00119-f005:**
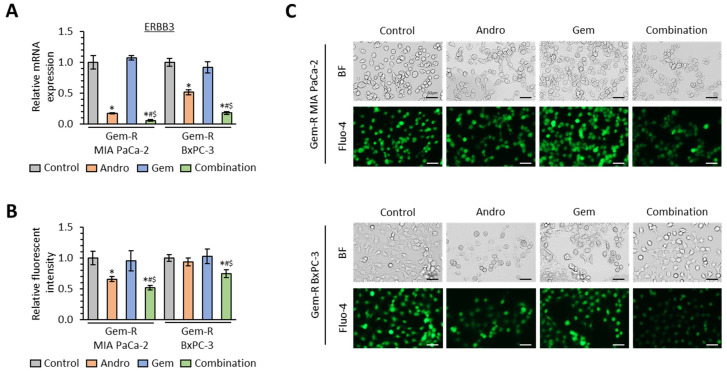
The combination of Andrographis and Gemcitabine down-regulates ERBB3 expression and decreases intracellular calcium concentration in Gemcitabine-resistant PDAC cells. (**A**) qRT-PCR analyses of ERBB3 in Gem-R PDAC cells following treatment with Andro, Gem, and their combination for 48 h. Relative expression was calculated using β-Actin mRNA expression as an internal control. The average (column) ± SD is indicated (* *p* < 0.05 vs. control, ^#^
*p* < 0.05 vs. Andro, ^$^
*p* < 0.05 vs. Gem). (**B**) The cellular fluorescence intensity value of Fluo-4 following treatment with Andro, Gem, and their combination in Gem-R PDAC cells. The average (column) ± SD is indicated (* *p* < 0.05 vs. control, ^#^
*p* < 0.05 vs. Andro, ^$^
*p* < 0.05 vs. Gem). (**C**) Representative images of Gem-R PDAC cells following treatment with Andro, Gem, and their combination using a bright-field or fluorescent microscope with a 494 nm/506 nm excitation filter under 160× magnification (scale bar = 50 μm). PDAC, pancreatic ductal adenocarcinoma; Gem-R, Gemcitabine-resistant; Andro, Andrographis; Gem, Gemcitabine; SD, standard deviation. BF, bright-field.

**Figure 6 biomedicines-11-00119-f006:**
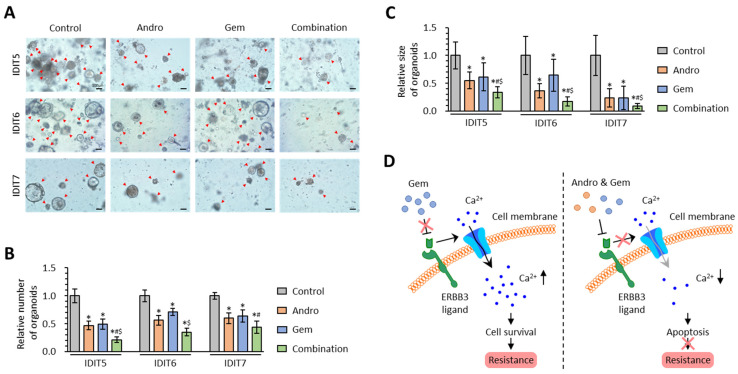
The combination of Andrographis and Gemcitabine effectively enhances anti-cancer activity in PDAC patient-derived organoid models. (**A**) Representative images of tumor organoids following treatment with Andro, Gem, and their combination. Scale bar = 100 μm (Magnification 100×). (**B**,**C**) Bar charts of relative number (**B**) and size (**C**) of tumor organoids following treatment with Andro, Gem, and their combination. The average (column) ± SD is indicated (* *p* < 0.05 vs. control, ^#^
*p* < 0.05 vs. Andro, ^$^
*p* < 0.05 vs. Gem). (**D**) A schematic illustration of Andro-mediated reversal of Gem resistance in PDAC. This illustration demonstrates Gem resistance through ERBB3 ligand-dependent calcium influx in PDAC cells (left) and the role of Andro in reversing Gem resistance through regulation of calcium signaling in PDAC cells (right). PDAC, pancreatic ductal adenocarcinoma; Gem-R, Gemcitabine-resistant; Andro, Andrographis; Gem, Gemcitabine; SD, standard deviation; Ca, calcium.

## Data Availability

Data are contained within the article and Appendix A.

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
