# Peer review of "Andrographis Reverses Gemcitabine Resistance through Regulation of ERBB3 and Calcium Signaling Pathway in Pancreatic Ductal Adenocarcinoma"

_biomedicines, 2023, doi:10.3390/biomedicines11010119_

Round 1

Reviewer 1 Report

1.Figure 3D, there are more than two blurred red line in images GEM-R(Gemcitabine-resistant) MIA PaCa-2 control 24hours, Gem 24 hours, Gem-R BxPC-3 control0 hour, 24hours, Combination 0 hours.

2.Figure 4C, the results from western blot images do not match the quantification data. Based on the western blot images shown by the authors in figure 4C, Andrographis (Andro) significantly increased the caspase-3 protein level in Gem-R MIA PaCa-2 compared with the control, while in the quantification data the authors showed no significance. Same issue for Gem-R BxPC-3 control vs. Combination western blot image. If the author counted combination has twice the caspase-3 protein level as the control, why did Andro, which showed a similar caspase-3 protein level compared to the combination from western blot images, while the authors showed there is no significant between the control and Andro in caspase-3 protein level in the quantification data?

3. Cleaved caspase-3 western blot results are needed for the protein samples from the experiment in figure4C.

4. The authors emphasized that Andro significantly enhanced the chemosensitivity to Gem in the current manuscript. Drug synergy testing is needed, and a synergy score should be calculated.

5. Based on the data shown by the authors from figure4, figure5 and figure 6, Andro alone seems to be sufficient to induce Gem-R PDAC cell line apoptosis, decrease intracellular calcium concentration via down-regulates ERBB3 expression, and impact PDAC organoid growth.   Then what’s the point to use GEM+Andro combination treatment to treat Gem-R PDAC, instead of just using Andro treatment alone, if it is already GEM resistant under the clinic setting?

Reviewer 2 Report

The article by Dr. Goel and the group elaborates on the role of Andrographis in regulating chemoresistance. Also, it sheds light on the mechanistic part describing the role of ERBB3 and the calcium signaling pathway in it. This a well-planned experimental manuscript that follows through the hypothesis of the work. Though few things need to be addressed before it is ready for acceptance. They are as follows:

1. Authors should add a model at the end of fig 6 describing the finding from this manuscript and how it matches with the existing literature. This will be a graphical summary of this work. 

2. Authors must discuss some of the current work which describes other players who are involved in gemcitabine resistance. for example- EGFR, NRF2, etc. Refer to PMID: 31911550, PMID: 27477511, and other relevant ones. This can be done by adding a few lines in the discussion part.

3. Also, the authors must discuss KRAS mutation in pancreatic cancer, which is involved in more than 90% of PDAC, and discuss whether this finding relates to oncogenic RAS. This can be discussed as one of the possible future studies. Refer to recently published work on this PMID: 33870211 and  PMID: 32209560.

Reviewer 3 Report

The authors explore the ability of andrographolide, the main bioactive diterpene contained in Andrographis paniculata, to reverse Gemcitabine resistance in pancreatic cancer. With this purpose, they incubated the cells with andrographolide at a concentration of 25 μg per ml.

Unfortunately, Panossian et al.

Panossian A, Hovhannisyan A, Mamikonyan G, Abrahamian H, Hambardzumyan E, Gabrielian E, Goukasova G, Wikman G and Wagner H, Pharmacokinetic and oral bioavailability of andrographolide from Andrographis paniculata fixed combination Kan Jang in rats and human, Phytomedicine, 2000, 7, 351-364

found that the maximum achievable concentration of the drug in humans and animals in  plasma is approximately 393 ng/ml. Prolonged and repetitive administration can elevate the concentration to 660 ng/ml.

Therefore, they used a concentration which is 400 to 600% higher than what can be reached in a clinical setting.

This data, which is not a minor data is not explained in the paper.

From an academic point of view, their research is perfectly valid, although it is not applicable to any possible bedside treatment.

It is required that the authors explain how their research performed at non-attainable concentrations outside the laboratory, can become a contribution to treat PDAC patients with gemcitabine resistance.

Remembering the Paracelsus principle a poison is only a matter of dose.

According to Richard et al. concentrations above 10 μg per ml can be cytotoxic.

Richard, E. J., Murugan, S., Bethapudi, B., Illuri, R., Mundkinajeddu, D., & Velusami, C. C. (2017). Is Andrographis paniculata extract and andrographolide anaphylactic?. Toxicology Reports4, 431-437.

The authors also forget to mention that andrographolide was found to posses easily ignored security issues in clinical application, such as nephrotoxicity and reproductive toxicity.

Zeng, B., Wei, A., Zhou, Q., Yuan, M., Lei, K., Liu, Y., ... & Ye, Q. (2022). Andrographolide: A review of its pharmacology, pharmacokinetics, toxicity and clinical trials and pharmaceutical researches. Phytotherapy Research36(1), 336-364.

Andrographolide also interferes with immune mechanism by impeding T-cell activation. This can be pro- or antitumoral.

Iruretagoyena MI, Tobar JA, Gonzalez PA, Sepulveda SE, Figueroa CA, Burgos RA, et al. Andrographolide interferes with T cell activation and reduces experimental autoimmune encephalomyelitis in the mouse. J Pharmacol Exp Ther. 2005;312(1):366–72. https://doi.org/10.1124/jpet.104.072512

This may be an important drawback in cancer treatment.

Another omission by the authors is the inhibitory activity of NF-kB which is a major antitumoral effect.

Tan WSD, Liao W, Zhou S, Wong WSF. Is there a future for andrographolide to be an anti-inflammatory drug? Deciphering its major mechanisms of action. Biochem Pharmacol. 2017;139:71–81. https://doi.org/10.1016/j.bcp.2017.03.024.

And inhibits the PI3K/Akt axis which is another important antitumoral activity

Andrographolide inhibits PI3K/AKT-dependent NOX2 and iNOS expression

Chern CM, Liou KT, Wang YH, Liao JF, Yen JC, Shen YC. Andrographolide inhibits PI3K/AKT-dependent NOX2 and iNOS expression protecting mice against hypoxia/ischemia-induced oxidative brain injury. Planta Med. 2011;77(15):1669–79. https://doi.org/10.1055/s-0030-1271019.

The authors center their research on the calcium blocking activities of ANDRO, however, they made no comparison with a drug that is classically used for resistance reversal such as verapamil, that happens to be a calcium channel blocker. In this regard is ANDRO better, worse or equal than verapamil?

They leave this issue without an answer.

They mention that there objective is limited:

"since the primary purpose of the present study was to evaluate Andro’s anti-cancer potential to enhance chemosensitivity to Gem in PDAC cells"  

This objective was reached.

Unfortunately, the unanswered questions that were explained above limits also the perspective that ANDRO may become a usable drug to target gemcitabine resistance.

Round 2

Reviewer 1 Report

Most of the concerns have been addressed.

Please make sure that figure 4C, cleaved caspase-3 western blot results are replaced by new images before publication. The authors cropped cleaved-casepase3 western blot results in the middle of the current image.

Author Response

  1. Please make sure that figure 4C, cleaved caspase-3 western blot results are replaced by new images before publication. The authors cropped cleaved-casepase3 western blot results in the middle of the current image.

Response: We would like to thank the reviewer for valuable suggestions to improve the manuscript. In this resubmission, we have replaced the new image of cleaved caspase-3 and have attempted to improve the overall appearance of Fig. 4C.

Reviewer 2 Report

All concerns have been addressed- ready for acceptance. 

Author Response

There were no comments from this reviewer

Reviewer 3 Report

No further comments

Author Response

(The authors gave the same response as above.)
